

# Compact bulk-machined electromagnets
# for quantum gas experiments

**Kevin Roux, Barbara Cilenti, Victor Helson, Hideki Konishi and Jean-Philippe Brantut⋆**

Institute of Physics, École Polytechnique Fédérale de Lausanne, 1015 Lausanne, Switzerland

⋆ jean-philippe.brantut@epfl.ch, http://lqg.epfl.ch

## Abstract

We present an electromagnet combining a large number of windings in a constrained volume with efficient cooling. It is based on bulk copper where a small pitch spiral is cut out and impregnated with epoxy, forming an ensemble which is then machined at will to maximize the use of the available volume. Water cooling is achieved in parallel by direct contact between coolant and the copper windings. A pair of such coils produces magnetic fields suitable for exploiting the broad Feshbach resonance of $^6$Li at $832.2$ G. It offers a compact and cost-effective alternative solution for quantum gas experiments.



# 1 Introduction

Strong, homogeneous magnetic fields of the order of a thousand Gauss are at the core of quantum gas experiments, from laser cooling and trapping [1] to the use of Feshbach resonances controlling the inter-atomic interactions [2]. As experimental apparatus become more and more complex, with more laser beams or high-aperture imaging systems to accommodate, space occupation as well as heat management constraints become more and more acute, calling for optimized and flexible electromagnet designs. Efforts in this direction have been reported in the past years, in particular novel designs for Zeeman slowers [3,4], Bitter electromagnets [5–7] or improved heat management methods [8]. More compact systems of magnetic traps use in-vacuum electromagnets [9–11], or atom chips [12,13], but those are not adapted to experiments requiring large homogeneous fields.

The common ground of these electromagnet concepts is the improvement over the widely used design based on wound copper wire (see for example [14]). This solution is privileged due to its robustness, and the use of hollow wire allows for water cooling with good contact between copper and the coolant. It suffers however from several drawbacks, the first being the need for coolant and electrical current to follow the same path, yielding a large coolant pressure drop across the magnet even for a moderate number of turns. Several circuits of coolant fluid are then needed to limit temperature inhomogeneities in the coil. Another drawback is the lack of flexibility and poor space occupation efficiency: hollow copper wires are typically several millimeters wide, restricting the number of windings in a given volume. In addition, the epoxy matrix and coolant circuit inlets and outlets limit the fraction of volume actually used for current carrying copper.

The Bitter-type configuration addresses some of these drawbacks [5–7], in particular it offers very good heat management thanks to the improved flow of coolant through the magnet, at the cost of requiring each winding to bear several coolant connections. Space occupation was addressed in a recent improvement where several layers are used to increase the number of windings [15]. Other alternatives include the use of mixed configurations where hollow copper is combined with bulk copper wires [16] in a single assembly, or a scheme with fully parallel cooling demonstrated in a few-windings coil directly machined from a bulk copper block [8].

In this article, we present electromagnets optimizing space occupation, offering very large shape flexibility while allowing to reach the high fields required for the use of Feshbach resonances in $^6$Li [17]. Our design is based on a bulk copper plate in which a spiral is cut by wire erosion [8], and impregnated with epoxy. The resulting ensemble is machinable using standard tools, allowing for carving out ridges to maximize space occupation or holes for electrical connections and clamping. As a result, the limited volume of a reentrant vacuum viewport accommodates 31 windings over a single 22 mm thick layer. We present the detailed fabrication procedure of the ensemble, as well as performance in terms of magnetic fields and heat management, demonstrating the suitability of this approach for quantum gas experiments.

# 2 Concept and design

The maximum magnetic field that a coil electromagnet can produce is limited by the geometric constraints on the coil dimensions and the total allowable power dissipation. In designing electromagnets for cold atoms experiments, the dimensions are constrained by the vacuum systems and necessary optical access, which limit the distance to the atoms and restricts the total volume available for the current carrying coil. In addition, the electric current is limited by practical constraints such as the diameter of cables, heating at the contacts and availability

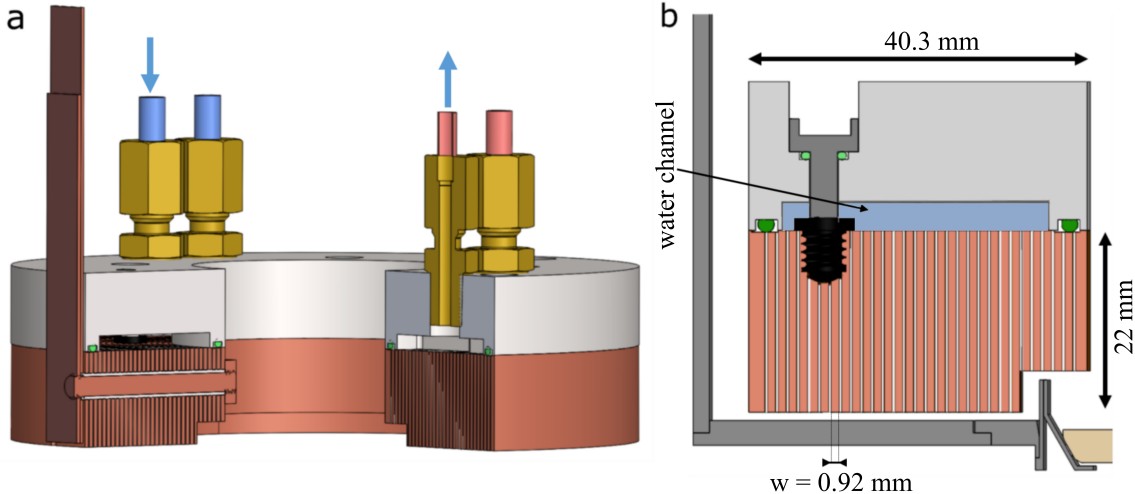

Figure 1: **a**: 3D Computer-assisted drawing of the electromagnet. The copper coil consists of horizontally stacked windings (orange), separated by epoxy layers. A PEEK[5] cap (light grey) creates a channel on top of the coil, rendered water tight by EPDM[6] o-rings (green), where cooling fluid is injected (light-blue arrows) and circulates in direct contact with the copper. At the bottom of the coil, a ridge is machined to fit the geometry of the reentrant viewport. A through hole allows for the electrical connection of the inner winding from the outside through the coil body. Electric current is injected and collected by a pair of flat cables running upward. **b**: Cut view of the electromagnet, showing blind threaded holes with PEEK inserts (black) hosting titanium screws to hold the cap and the coil together. The resulting assembly optimally fits the available space in the reentrant viewport with current carrying copper.

of power supplies.

In our design, the electromagnet has to fit in the limited volume of a reentrant vacuum viewport, while allowing for the use of high-resolution optics. At the same time, a pair of such electromagnets shall achieve magnetic fields of hundreds of Gauss several centimeters below the coil, in order to reach Feshbach resonances in $^6$Li atoms, with a current limited to 440 A by the power supply. We meet these requirements using a coil comprising 31 turns horizontally stacked in a single 22 mm thick layer, depicted in figure 1, with a large aspect ratio of 23.9 for an individual winding. The inner and outer radii are 32 and 72 mm respectively.

With the geometry fixed, the performance is limited by power dissipation, which is achieved by forced convection with cold water. Three different processes determine cooling efficiency and thus the temperature distribution within the electromagnet: (i) heat conduction within the bulk of the coil, measured by the temperature difference across the copper coil $\Delta T_c$ (ii) heat transfer at the coil-coolant interface, measured by the temperature difference between the coil and the coolant $\Delta T_w$ and (iii) coolant flux through the assembly, measured by the coolant temperature difference $\Delta T_f$ between the inlet and outlet. The regime $\Delta T_f \gg \Delta T_c + \Delta T_w$ corresponds to *flux-limited* cooling, where the coolant and magnet are thermalized and heat removal is limited by the flow, while $\Delta T_f \ll \Delta T_c + \Delta T_w$ corresponds to a *transfer-limited* regime where heat transfer between the coil and the coolant limits the cooling.

To guide our design, we estimate the parameter dependence of these three temperature differences for our horizontal stack configuration, where the electromagnet is cooled by cold water in direct contact with the upper surface of the coil. For a given current $I$ in the windings,

neglecting lateral heat flow between windings, the temperature difference between the top and bottom of the coil reads

$$\Delta T_c = \frac{I^2}{w^2}\frac{\rho_{\text{Cu}}}{2\lambda_{\text{Cu}}},\tag{1}$$

with $\rho_{\text{Cu}}$ and $\lambda_{\text{Cu}}$ the electrical resistivity and heat conductivity of copper, respectively, and $w$ the width of one winding (see Appendix A for the derivation). For a current of 400 A and $w = 1$ mm, we get $\Delta T_c = 3.4$ K, showing that gradients within the coil will be minimal in spite of the very large aspect ratio for individual windings.

The heat transfer at the copper-water interface depends on the nature of the flow. For a typical total flow rate in the coil 0.23 l·s$^{-1}$, a simple estimate based on the hydraulic diameter of the duct yields turbulent flow with a Reynolds number of $\sim 6.4 \cdot 10^3$ [18][1]. We deduce a heat transfer coefficient $h_w \sim 5 \cdot 10^3$ W·m$^{-2}$K$^{-1}$ and thus a temperature difference (see Appendix A for derivation)

$$\Delta T_w = I^2 \frac{\rho_{\text{Cu}}}{H w^2 h_w},\tag{2}$$

with $H$ the coil thickness. For our design, we evaluate the ratio $\Delta T_c / \Delta T_w \sim 0.14$, which indicates that the interface is the limiting factor in the cooling [2].

Last, we can easily estimate the increase of water temperature across the coil from simple energy conservation considerations, yielding

$$\Delta T_f = I^2 \frac{R}{C_w Q},\tag{3}$$

where $R$ is the total electrical resistance of the coil, $Q$ is the coolant flux and $C_w$ is the volumetric heat capacity of water. For our system allowing large water fluxes of 0.23 l·s$^{-1}$, we estimate $\Delta T_f \sim 0.7$ K for $I = 400$ A.

This shows that even with an extreme aspect ratio for the coil windings, heat diffuses efficiently through the coil up to the water-copper interface, and that we operate well in the transfer-limited regime in contrast to most designs [7,14]. Operating in the transfer-limited regime has the advantage that heat removal is less sensitive to the water flux since the flux enters only through the heat transfer coefficient with a sub-linear dependence [18].

## 3 Manufacturing and assembly

Manufacturing of the electromagnet followed the concept of [8], where a coil is carved out from bulk copper rather than wound out of wires. As described below, a major advantage of this concept is that the thick body of the coils forms a single, rigid ensemble, which can then be shaped using standard lathe and milling machines. As long as machining does not significantly affect the windings, the final shape can fulfill a wide variety of space constraints, with negligible effects on the magnetic field distribution. We used this capability in order to carve out tapped holes for holding the cap, a through hole carrying current through the coil body from the inner to the outer part and an edge in order to fit the exact geometry of the reentrant viewport, as can be seen in figure 1.

---

[1]The large aspect ratio and short length of the duct would call for a more detailed analysis of the flow, beyond the scope of the present work

[2]Within the fluid, heat is carried predominantly by convection rather than conduction, as indicated by a Nusselt number of $\sim 70$ [18]

## 3.1 Coil body

Manufacturing started from a 24 mm thick oxygen-free copper plate. Wire-erosion machining creates a spiral with a pitch of 1.3 mm, leaving a gap between consecutive windings of 0.38 mm. Fiber-glass reinforced plastic spacers were inserted between successive windings to preserve uniform spacing and ensure electrical insulation. After cleaning and drying, the spiral was impregnated with epoxy glue. We used commercial low viscosity, room-temperature curing epoxy[3], loaded with 10% of 0.25 µm fiber glass flake and 30% aluminium nitride (AlN) powder[4]. Preliminary tests showed that cracks can develop in the inter-windings regions due to thermal stress when using high temperature curing expoxy, hence the use of a fiber glass load to reinforce the structure. We also found that the use of AlN load improves the heat dissipation from the inner and outer windings of the coils, which are not in direct contact with water.

The impregnated coil was then machined with a lathe in order to remove the excess glue and expose the copper on the coil facet to be water cooled. The resulting surface quality is high, as can be seen in figure 2, demonstrating the efficiency of machining of the composite assembly. Further machining could be used to roughen the surface on purpose, favoring turbulent flow to improve heat transfer. The assembly consists in more than 70% copper in volume. Blind threaded holes were machined on the top surface to fit screws pressing the cap against the coil. A transverse through-hole was drilled and a fiber-glass insulating tube glued inside to accommodate a 5.5 mm diameter copper screw, carrying current from the inner to outer winding (see figure 1).

## 3.2 Electromagnet assembly

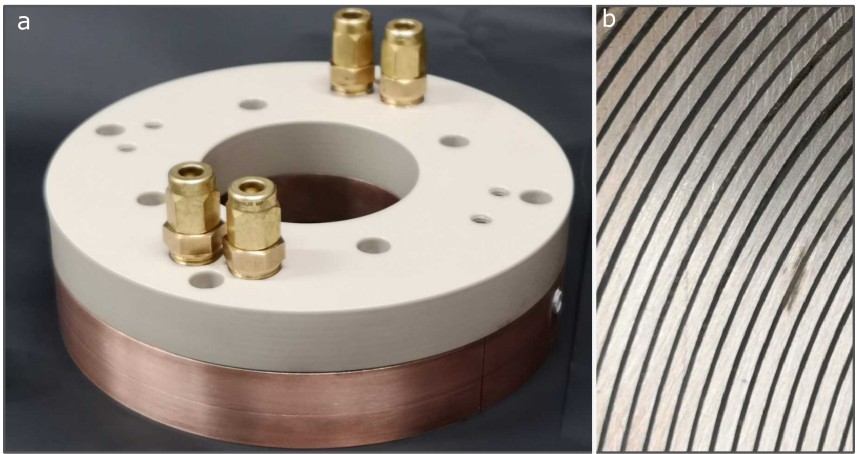

Figure 2: **a**: Photograph of the assembled electromagnet, showing the PEEK cap attached to the coil body, with coolant connections from the top. **b**: Close view of the top surface of the coil after glueing and machining on a standard lathe, showing the successive windings separated by the electrically insulating epoxy. The width of a copper winding is 0.92 mm.

As shown in figure 1, the electromagnet mainly consists of the coil and the plastic cap creating the water cooling channel and holding the ensemble together. The cap is a U-shaped

---

[3]Sicomin SR1710/SD7820
[4]Sigma Aldritch 10µm powder, ≥ 98% purity

PEEK[5] part forming a 3 mm high channel on the top surface of the coil, with a cross-sectional area of 102 mm$^2$. To increase the water flux, the channel is split in two half-circles with separate inlets. The cap is attached to the coil body by 8 titanium screws fitting in threaded holes machined on the top surface of the coil, with PEEK inserts for electrical isolation. Titanium offers high stiffness, is non-magnetic, and is more cathodic than copper which reduces the risk of galvanic corrosion. PEEK inserts fit into the threaded holes to electrically insulate the coil from the screws, and holes are rendered leak tight using EPDM[6] O-rings. Two long O-rings pressed in the inner and outer sides of the cap ensure that the ensemble is leak-tight[7] up to 4 bars overpressure.

Electric current is injected and collected via two copper plates separated by a thin electrical insulation, minimizing stray magnetic fields (see figure 1). Current is collected from the inner side via a copper screw going through the coil itself. This configuration, made possible by our machinable coil concept, further optimizes the use of space and minimizes stray fields since it does not require an electrical wire running from the inner part of the coil[8].

The resulting system, shown in figure 2, is highly compact. The cap is used to suspend the ensemble such that all the available space of the reentrant viewport is occupied by current carrying copper, maintained as close as possible to the vacuum chamber surface but mechanically disconnected from it.

## 4 Performance

### 4.1 Electromagnetic properties

We first characterized the magnetic field distribution with a DC current and the electromagnet in the steady state. The axial component of the magnetic field $B_z$ produced by the electromagnet was measured with a uniaxial Hall probe[9] as a function of position. The results are presented in figure 3. For comparison, we performed numerical simulations using the Radia package [19] considering homogeneous current distributions inside the coil body. These are presented as solid lines in the figure, showing very good agreement with the measurements.

We measured a peak magnetic field at the expected position of the atoms, 52.2 mm below the mean coil position, of 1.28 G·A$^{-1}$, implying that a pair of such coils separated by 104.4 mm will yield a field of 832.2 G, the position of the broad Feshbach resonance of $^6$Li [17], at a current of 325 A. Our simulation predicts the field distribution for this pair of coils, as shown in figure 3c. At the position of the atoms, the field varies quadratically with a curvature along the axial direction of 0.11 G·A$^{-1}$cm$^{-2}$. The combination of a finite curvature, yielding magnetic trapping of high-field seeking atoms along the radial directions, with a dipole trap providing confinement along the axial direction is a standard configuration for cold $^6$Li atoms experiments [20]. Variations of magnetic fields across a typical atomic cloud close to the Feshbach resonance are negligible compared with the 262 G width of the resonance. The simulation also showed that the current carried in the through hole from the inner to outer winding, which breaks the cylindrical symmetry of the ensemble, results in negligible disturbance in the field distribution.

We measured the inductance of the coil, which characterizes its dynamical response. Its impedance was measured as a function of frequency between 0.16 Hz and 320 Hz. We fitted a DC resistance of 10.4(10) mΩ and an inductance of 116(2) μH, in reasonable agreement

---

[5] Polyether ether ketone
[6] Ethylene Propylene Diene Methylene rubber. EPDM 70, Shore A, 1.5 mm cord, 6 mm diameter
[7] We use Curil T (Elring) grease for matching the O-rings with the surface.
[8] We use Chemtronics CW7100 grease for contact between the copper bolt and the windings
[9] Lakeshore 425 with HMNA-1904-VR probe

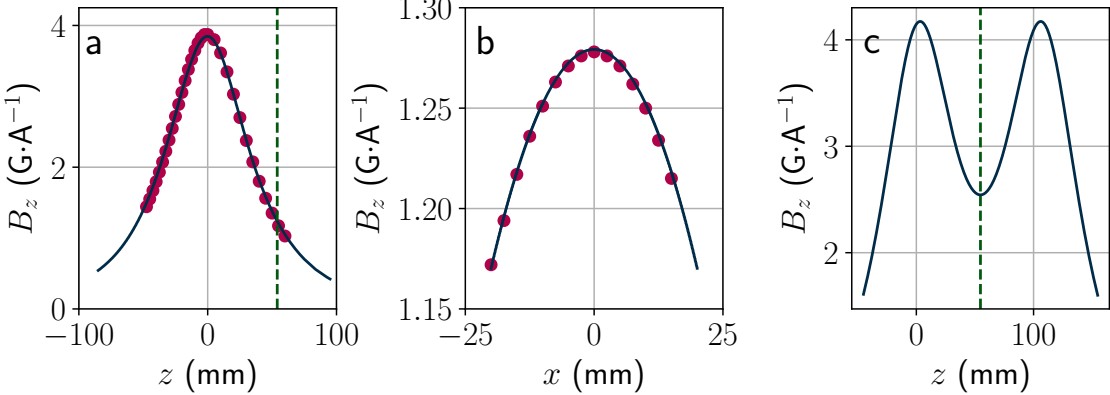

Figure 3: Axial component of the magnetic field created by one electromagnet as a function of **a**: the position $z$ along the coil axis, and **b**: the position $x$ in the plane of the atoms located 52.2 mm from the center of the coil (dashed vertical line on **a**). The red points are measurements, and the solid lines represent simulations using the Radia package. **c**: Simulated axial magnetic field created by a pair of identical coils separated by 104.4 mm. The dashed vertical line represents the expected position of the atoms.

with the magnetic field simulation yielding $94\,\mu$H and with the expected intrinsic resistance of $8.7\,$mΩ. This is comparable with the results obtained for Bitter electromagnets with a similar geometry and number of turns [7].

## 4.2   Heat management

We tested the thermal performance of the electromagnet by measuring the temperature at different points of the assembly during operation. We first imposed a fixed cooling water flux of $0.23\,$l·s$^{-1}$ with a water temperature of 17.5$^{\rm o}$C. The steady state temperature as a function of current is presented in figure 4a at various locations in the system. The overall temperature of the coil body, except for the first inner and outer winding, measured on the bottom side of the coil, remains below 30$^{\rm o}$C. This confirms our expectations that heat is efficiently removed by water running over the edges of the coil windings. The heating rate of the coil body is $9.4\,$K·kW$^{-1}$.

The water temperature increase across the system was very low, and remained lower than at any other point in the electromagnet, confirming that it operates in the transfer-limited regime. We observed that the temperature is larger at the inner and outer windings than in the bulk of the ensemble, because these windings are not in direct contact with water, as can be observed in figure 1. This results from the design choice to privilege the number of windings over the thermal homogeneity[10]. Operating the electromagnet in realistic experimental conditions, with 350 A current and a duty cycle of 30%, we observed that the highest temperature was reached in the inner winding, with a steady-state value of 34$^{\rm o}$C, compatible with routine operations in the laboratory.

We then measured the temperature variations with coolant flux. Thanks to the wide section of the coolant circuit, the water flux was only limited by the diameter of the inlet pipes, allowing for total fluxes up to $0.261\,$l·s$^{-1}$ for a moderate water pressure drop of 3.5 bars across the electromagnet. The heating of the coil body is reduced upon increasing the water flux,

---

[10]Using the same machining capabilities, our concept allows for the use of a variable pitch spiral ensuring direct water cooling of each winding. In the present case, this would have reduced the number of windings by $\sim 4$

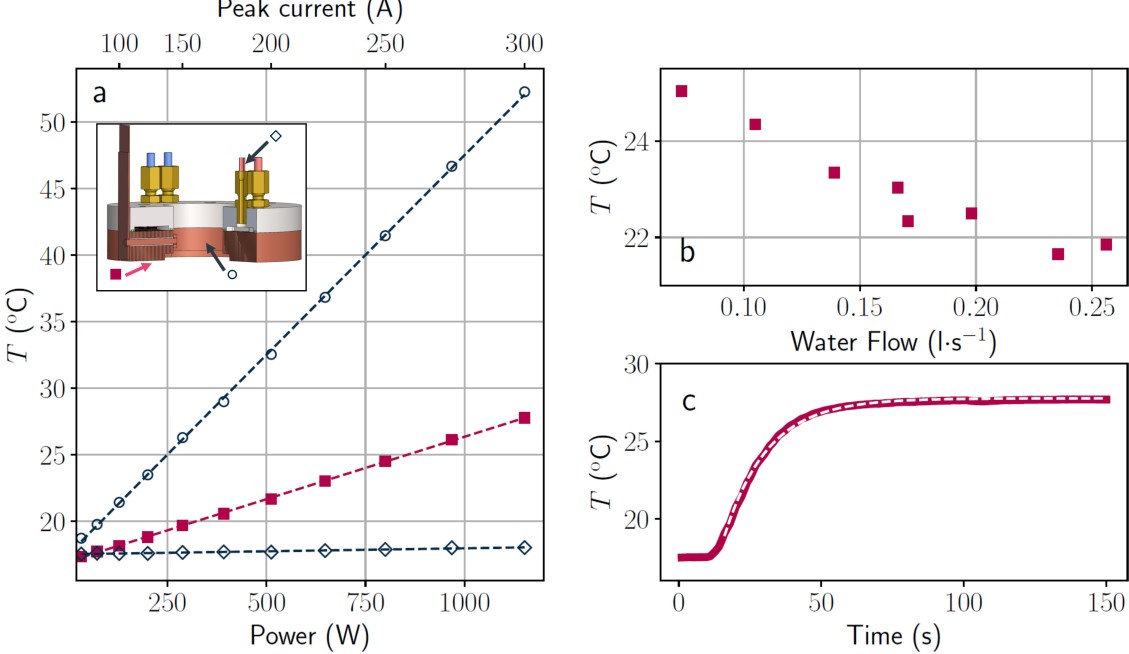

Figure 4: **a**: Steady state temperature of the electromagnet as a function of electrical power dissipated with a total coolant flux of $0.23\,\mathrm{l\cdot s^{-1}}$, measured at the inner winding (circles), and bottom surface (squares), and temperature of the water exiting the magnet (diamonds). The dashed lines are linear fit to the data, yielding heating rates of 0.45, 9.4 and $30\,\mathrm{K\cdot kW^{-1}}$ for the water, coil body and inner winding, respectively. **b**: Temperature of the coil body as a function of the total water flux in the coil, for an average electrical power of 512 W. **c**: Time evolution of the temperature of the coil body following a sudden switch-on of 300 A. The thick red line represents measurements and the dashed line is an exponential fit yielding a timescale of 15.26 s.

which is expected due to the dependence of the heat transfer coefficient on velocity in the turbulent regime.

Finally, we measured the dynamics of the thermalization of the coils, monitoring the temperature after switching on the electrical current to 300 A, with a water flux of $0.23\,\mathrm{l\cdot s^{-1}}$. The result is shown in figure 4c. The evolution is well fitted by a single exponential, which timescale $\tau$ allows for an estimate of the average copper-water heat transfer coefficient $h$: energy balance considerations, supposing a homogeneous temperature within the coil, yields $\tau \sim wC_{\mathrm{Cu}}/h = 15.3\,\mathrm{s}$, with $C_{\mathrm{Cu}}$ the specific heat of copper, from which we obtain $h \sim 5 \cdot 10^3\,\mathrm{W\cdot m^{-2}K^{-1}}$, in agreement with our initial expectations.

This short thermalization time is favorable since thermal expansion of the coils changes the magnetic field, such that accurate calibration requires a steady-state situation. For our geometry, we estimate that changes of the coil radius with temperature produce relative variations of the magnetic fields of $\sim 1.3 \cdot 10^{-5}\,\mathrm{K^{-1}}$. Thermal effects upon varying the coil temperature will be comparable with the effects of finite accuracy of standard power supplies.

## 5 Conclusions

We have presented a compact and flexible electromagnet concept adapted to quantum gas experiments requiring both large homogeneous magnetic fields and minimal space occupa-

tion. The overall costs are minimal, limited to that of the raw copper, inexpensive epoxy glue, and a home made custom plastic mold for the gluing operation. Cutting the spiral required $\sim 40$ hours on the wire erosion machine of our institute's mechanical workshop, running fully automatically.

Our experimental system is constrained by the need for high numerical aperture together with the accommodation of a large, in-vacuum experimental platform, leaving only limited space for the main electromagnets. Thanks to the compactness of the main electromagnets, the reentrant viewports of our experimental setup can also accommodate 4 pairs of compensation coils in the cloverleaf configuration [21], allowing for moving the saddle point of the field in the plane of the coil. This possibility will allow for precise positioning of the atomic cloud with respect to the experimental platform. It will also allow for transport measurements with lithium quantum gases in the two-terminal configuration [22].

# Acknowledgements

We acknowledge the technical assistance of Claude Amendola, Olivier Haldimann, Gilles Grandjean, Philippe Zuercher and Damien Fasel.

**Funding information**    We acknowledge funding from the ERC project DECCA (Project No. 714309), the Sandoz Family Foundation-Monique de Meuron program for Academic Promotion and EPFL.

# A    Temperature gradients across the electromagnet

## A.1    Conduction through the coil body

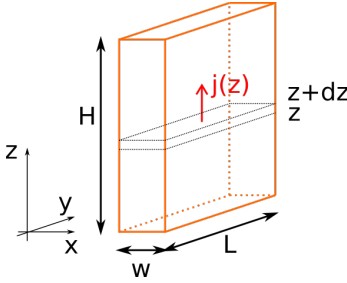

Figure 5: Calculation of the temperature profile within one winding, based on energy balance within a slice of thickness $dz$.

We consider one winding of the coil as a prism with rectangular section of width $w$ and height $H$, with a total length $L$, as illustrated in figure 5. With the notations of the main text, the total power dissipated is

$$P = \rho_{\text{Cu}} \frac{L}{Hw} I^2 = pLHw \,, \tag{4}$$

where we introduced the power dissipated per unit volume $p$.

To calculate the temperature profile we neglect heat flowing on the sides, and consider the temperature as homogeneous in the $x - y$ plane. In the steady state, the energy contained in

the slice between $z$ and $z + dz$ is constant, imposing

$$\frac{dj}{dz} = p \, , \tag{5}$$

where $j$ is the heat flux through unit area. With $j(0) = 0$ we get $j(z) = pz$. Fourier's law then relates the temperature profile $T(z)$ to the heat flux

$$\lambda_{\text{Cu}} \frac{dT}{dz} = j(z) \, , \tag{6}$$

which yields

$$T(z) = T(0) + \frac{pz^2}{2\lambda_{\text{Cu}}} = T(0) + I^2 \frac{\rho_{\text{Cu}}}{2\lambda_{\text{Cu}} w^2} \frac{z^2}{H^2} \, . \tag{7}$$

Evaluating this expression for $z = H$ yields equation 1. The dependence on $H$ has dropped since upon varying the thickness at fixed total current, the increase of heat resistance is balanced by the decrease of total dissipated power.

## A.2   Coil-water interface

We consider the temperature at the coil-coolant interface in the presence of forced convection. In steady state, the continuity of the heat flux at the interface imposes that

$$h_w(T(H) - T_w) = Hp \, , \tag{8}$$

where $T_w$ and $T(H)$ are the temperatures of the coolant and coil at height $H$ calculated above, $h_w$ is the heat transfer coefficient at the interface, and $p$ is the power dissipation per unit volume within the coil. We obtain

$$\Delta T_w = I^2 \frac{\rho_{\text{Cu}}}{H w^2 h_w} \, . \tag{9}$$

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
