# Peer review of "Compact bulk-machined electromagnets for quantum gas experiments"

_SciPost Physics, doi:SciPost Phys. 6, 048 (2019)_

## Round 1 · Referee Report · Anonymous (Referee 1) · 2019-2-22

Strengths
2-Well written and laid out manuscript.
Weaknesses
1-Some technical information is not provided with sufficient detail (e.g. the composition of some materials, the derivation of Equation 1 - see requested changes). 2-Perhaps some of the figures could be more informative. 3. There could be more of a discussion regarding how the coils will perform in their designated task - producing a stable and uniform magnetic field across a degenerate quantum gas of Li atoms. It is unclear what expected variation in scattering length - spatially and temporally - these coils are likely to produce under experimental conditions.
Report
Overall, this is a well written paper reporting on an interesting approach to the technical design of a ubiquitous component (magnetic coils) in cold atoms experiments. While their approach may not be entirely novel and does not yield any new physics, it is a different technical solution that appears to fulfill the design constraints well. Given the number of magnetic field coils used in cold atoms (and other) experiments worldwide, the paper will thus be of interest to the experimental cold atoms community. I therefore think that it is worthy of publication, subject to the following suggested changes.
Requested changes
1- Fig. 1 should have a few more parts labeled for clarity (either in the caption or on the image), and the materials that components such as the black insulators, green dots, plastic caps are made from should be mentioned. Indicating the physical meaning of some of the variables (w, H etc), on this Fig might also be helpful.
2- It would be helpful to provide some more information of where equation (1) comes from. At the minimum a reference is needed, although some more information in the text might be helpful, as parts of it are based on assumptions (e.g. why does the distance which $\Delta T$ is over not come into the equation?).
3 - Similarly, I feel that the general reader would benefit if a comment and/or reference could be provided to expand on the estimate of a Nusselt number of ~70.
4 - What is the expected field variation across a typical atomic cloud size that will be achieved with these coils, and what variation in scattering length will this lead to at some typical FB resonance B field values? From some very rough estimates looking at Fig. 3, it seems like in the z direction it might be non-negligible. Maybe of order 1G across a 100um cloud size? A couple of sentences discussing this would be nice. What about temporal variations due to thermal heating of the coils?
5 - Why were the screws made of titanium? Was it for the low magnetisation? Is this use of a dissimilar metal likely to cause corrosion problems due to the relatively large electrode potential difference between Ti and Cu?
6 - What is the cooling water temperature, and is it actively maintained at a precise temperature?
7 - While the English throughout the paper is generally at a high standard, there are a couple of minor errors I noticed including:
- First sentence of last paragraph of section 2: missing "an" before "extreme".
- Last sentence section 3.1: should be "...a fiber-glass...".
- First sentence of section 3.2: "in" should be "of".
- Last paragraph of section 4.2: "Last" should be "Finally", plus the last sentence is extremely long and should ideally be broken up for readability.
We thank the referee for his/her positive appreciation of our work. The new version of the paper adresses the detailed comments and requested changes. See the list of changes for detailed answers to each of the requests.

Author: Jean-Philippe Brantut on 2019-03-18 [id 468]
(in reply to Report 2 by Ryan Thomas on 2019-02-26)''I have two main concerns with the manuscript. The first is that their explanation of their design considerations is unclear. Equation 1, which is used to calculate the top-to-bottom temperature gradient on the coil, is presented without a reference or derivation. Equation 2 is similarly presented with little explanation, and I find its purpose vague and unclear.''
We thank the referee for pointing out the lack of clarity. This part has been entirely rewritten. In addition we have written an appendix presenting the derivation of the equations. The purpose is now clarified, by relating them to the various regimes of heat dissipation in the coil.
''On a similar note, the authors include a photograph of the top surface of their coil to showcase its surface quality. It is, however, unclear why this is important. Would a roughly cut top surface significantly degrade the heat management?''
The purpose of the picture is to demonstrate the machinability of the ensemble: a straightforward operation on a lathe produces a nearly perfect surface. This is not related to the cooling efficiency. As we now write in the paper, further machining could be used to produce roughness on purpose, that would favor cooling by favoring turbulence.
''My second concern is that the discussion of the magnetic field generated by the coil is limited. For instance, the authors state that a pair of coils is to be used for generating homogeneous fields at the location of their sample of atoms; however, the arrangement of this pair of coils is not stated. Is it supposed to be in a Helmholtz configuration? What is the homogeneity of the axial field at the location of the atoms?''
Indeed, this information was not explicitly given. We now include a new figure with the calculated field profile for a pair of coils, and the field curvature expected at the position of the atoms is given in the text.
''How fast can the magnetic field be changed: i.e., what is the dynamical response of the coils?''
We now describe the results of a measurement of the inductance of the coil, providing the necessary information on the dynamics of the field.
''How much does the magnetic field change with changes in the temperature of the coil?''
We thank the referee for pointing out this interesting question. We have included a paragraph with an estimate of the change of the relative change of magnetic field with thermal expansion of the coil, which is of the order of 10^{-5} per Kelvin.
''As a last comment: this manuscript would benefit from further proofreading, as some of the phrasing and grammar negatively impacts the clarity of the article.''
We have corrected several typos and reformulated sentences where english could be improved.

---

## Round 1 · Referee Report · Ryan Thomas (Referee 2) · 2019-2-26

Strengths
1) The authors present an impressively compact design with excellent heat management that will be of interest to other experimentalists. 2) The description of the manufacturing process is comprehensive and easy-to-follow. 3) The authors have made an effort to explain their design considerations, which means the design will be more easily adapted by other scientists.
Weaknesses
1) Phrasing and grammar impedes comprehension of the manuscript 2) Explanation of design considerations is somewhat confusing in terms of heat management concerns. Equations 1 and 2 have little accompanying justification or explanation, so their accuracy and relevance are hard to understand. 3) Discussions of the magnetic field itself are lacking in regards to the actual deployment of a pair of coils. For instance, the manuscript lacks a discussion of the axial field homogeneity for a pair of coils in the configuration that they will use in their experiment.
Report
In general, I think that this manuscript will be of the most interest to experimentalists who are either setting up a new experiment or investigating improvements to an existing apparatus. In particular, the authors' descriptions of how the electromagnet was manufactured and assembled are detailed enough to allow someone else to build a similar device and to avoid pitfalls that the authors clearly encountered. The authors' measurements make a clear case that their design is effective at managing heat for reasonable pressure heads. Although their design was necessitated by the limited space into which they could put their magnets, a similar design methodology could be used for ultracold atom systems that use small, rectangular 'science cells' for their experiments. In such a situation, the improved heat management would be an asset, although other systems (such as in Ref 7) might be a better option. One measurement which is not reported but would be of interest to other experimentalists is the frequency response of the coils. As the authors do not report this, it seems that they are unconcerned with the dynamical response, but others who wish to implement this design may want an estimate of how fast the field can be changed.
I have two main concerns with the manuscript. The first is that their explanation of their design considerations is unclear. Equation 1, which is used to calculate the top-to-bottom temperature gradient on the coil, is presented without a reference or derivation. Equation 2 is similarly presented with little explanation, and I find its purpose vague and unclear. On a similar note, the authors include a photograph of the top surface of their coil to showcase its surface quality. It is, however, unclear why this is important. Would a roughly cut top surface significantly degrade the heat management?
My second concern is that the discussion of the magnetic field generated by the coil is limited. For instance, the authors state that a pair of coils is to be used for generating homogeneous fields at the location of their sample of atoms; however, the arrangement of this pair of coils is not stated. Is it supposed to be in a Helmholtz configuration? What is the homogeneity of the axial field at the location of the atoms? How fast can the magnetic field be changed: i.e., what is the dynamical response of the coils? How much does the magnetic field change with changes in the temperature of the coil?
As a last comment: this manuscript would benefit from further proofreading, as some of the phrasing and grammar negatively impacts the clarity of the article.
Overall, I think that this manuscript is of sufficient interest to those in the field of ultracold atomic physics, and to those outside, to be published in SciPost Physics.
Requested changes
1) In the second and third paragraphs of the introduction the authors describe so-called parallel coolant flow as being problematic for hollow wire coils but as being a benefit for Bitter-type magnets. The authors need to clarify what is meant by 'parallel coolant flow' and revisit these descriptions to ensure that they are consistent. 2) In the second paragraph of the 'Concept and design' section, the authors state that they will use a pair of their coils to generate the large magnetic fields used for accessing the lithium Feshbach resonance. It would be useful to the reader to know what the configuration of this pair of coils will be. 3) Figure 1 could be improved by the use of additional colours to indicate where epoxy is used and where the water flows in the coils. On my first reading this was very confusing. 4) The last three paragraphs of the 'Concept and design' section need to be revisited. Equation 1 has no reference or derivation despite its clear importance to the manuscript. It may also be incorrect: my quick derivation indicates that their expression for the temperature difference is missing a factor of 3. Similarly, the importance of equation 2 is not clearly explained. In general, I found this section to be hard to follow. 5) The importance of the surface quality, as shown in Figure 2b, needs to be explained. Additionally, Figure 2b does not necessarily show well the surface finish of the coils. 6) In section 4.1, the authors should replace the word 'vertical' with 'axial' as being a more generally applicable description. 7) Also in section 4.1, the authors should discuss, measure, or calculate the field distribution resulting from a pair of coils in their desired configuration. In particular, the axial field homogeneity would be of interest to readers. 8) A discussion of the dynamical response of the coils should be included, as well as the response of the magnetic field to temperature changes. 9) In the conclusion, the authors say that making the coil took 40 hours on a wire-erosion machine, and in the abstract they say that their coil is a 'cost-effective' solution. Other institutions without a wire-erosion machine and trained machinist would be forced to have this made off-site, and it is not clear if making the part in this way would still be cost-effective. An estimate of the cost (I imagine their machinist would be able to provide this) would be very useful to readers. 10) The last sentence of the conclusion contains the confusing phrase '..a feature which revealed crucial in transport measurements...' and should be fixed. 11) References 3 and 8 have the full URL for the DOI in the DOI field, as opposed to just the DOI. This means the link does not function. Reference 21 does not have a DOI, and one should be included if it exists.

---

## Round 2 · Referee Report · Anonymous (Referee 3) · 2019-3-19

Report

I have read the updated manuscript and am happy that the authors have satisfactorily addressed all referee suggestions. I would recommend the paper be published as is.

I did pick up one minor typo in the caption of Fig 1, which should read "The copper coil consists of horizontally...".

  • validity: high
  • significance: good
  • originality: good
  • clarity: good
  • formatting: excellent
  • grammar: excellent

Author:  Jean-Philippe Brantut  on 2019-03-21  [id 470]

(in reply to Report 1 on 2019-03-19)
Category:
answer to question

We thank the referee for his careful reading the revised manuscript. A new version has been submitted with the typo corrected.

---

## Round 2 · Referee Report · Ryan Thomas (Referee 2) · 2019-3-20

Report

The authors have satisfactorily addressed all of my comments and questions regarding their manuscript. In particular, section 2 (Concept and Design) is much clearer.

Aside from some grammar and spelling issues, some of which I have detailed in "requested changes", the only remaining comment I have is that their coils are not in the Helmholtz configuration. In the interest of precision, the authors should refrain from using this terminology. It appears only in the caption for Figure 3 and should be easy to remove this reference.

I recommend that this paper be accepted.

Requested changes

1) Remove the reference to "Helmholtz configuration" in the caption for Figure 3, as the coil pair is too far apart to be in the Helmholtz configuration.

2) Section 4, second paragraph: The discussion of the finite curvature is confusing. The sentence starting with "This finite curvature..." refers to the previous sentence about the axial curvature but then states that high-field seeking atoms will be confined along radial directions. The authors should be more precise about which curvature they are referring to, because high-field seeking atoms will indeed be trapped radially but they will be anti-trapped axially.

3) In the last sentence of the abstract, it should be "gas" and not "gases"

4) Section 1, paragraph 2: example is misspelled as "exemple"

5) Section 1, paragraph 2, last sentence: "limit" instead of "limits"

6) Section 2, equation 2: while the variable H is defined in the appendix, it is useful for it to be defined in the main text as well.

7) Section 3.2, paragraph 1: In the first sentence, the phrase "consists in the coil" should be "consists of the coil"

8) In the next sentence, it should be "cross-sectional area", not "cross section area"

9) Section 4.2, paragraph 1: In the last sentence, the value of the heating rate appears at the very top of page 8 and can be easily missed when reading. In terms of formatting, Figure 4 should be fixed at the top of the page so that this value is not missed.

10) Section 4.2, paragraph 2: In the second sentence, it should be "We observed that the temperature..." instead of "We observed that temperature".

  • validity: -
  • significance: -
  • originality: -
  • clarity: -
  • formatting: -
  • grammar: -

Author:  Jean-Philippe Brantut  on 2019-03-21  [id 469]

(in reply to Report 2 by Ryan Thomas on 2019-03-20)
Category:
answer to question

We thank the referee for his careful reading of the updated manuscript. A revised version has been submitted, including the requested corrections. The description of the coils configuration as 'Helmholtz' has been removed to avoid possible confusions.

---

## Round 2 · Author Response

We are pleased to submit a revised version of our paper on Bulk-machined electromagnets for quantum gas experiments. We thank both referees for their careful reading of the paper and their constructive remarks. We are very pleased that both referees consider that our work is of sufficient interest for publication in SciPost Physics.

The revised version addresses all the criticism of the referees. In particular, the points raised by both referees concerning the particular geometry used in the actually implementation, as well as the derivation and motivation for the equations (1) and (2) are clarified in this new version.

---

## Round 2 · List of Changes

Changes requested in report 2:

1 - We have removed ‘parallel’ from the introduction since it was indeed used in an ambiguous way.
2 - This is now explicitly stated in the text.
3 - We have included more details in the figure. The caption has been expanded in order to provide a more detailed description of the figure.
4 - We have rewritten this paragraph. The detailed derivation of the various equations are now presented in the new appendix, which confirms that the result presented is correct.
5 - We have added a sentence to clarify the purpose of the figure, and the relationship with cooling efficiency.
6 - We have followed this recommendation.
7 - The new figure 3 now presents the field distribution for the pair of coils as implemented in the experiment, and numbers for the field curvature are stated in the text.
8 - The measured inductance is of the coil is now stated, as well as an estimate of the temperature sensitivity of the magnetic field.
9 - This is an interesting question. Wire erosion machines are not uncommon in university workshops. However, we did not mention explicit costs because (i) the machining cost will strongly depend on the actual capabilities of the external company, in particular the experience it has with cutting thin wires without them twisting and breaking the wire, (ii) the actual cost of manufacturing for our system is strongly influenced by the high labor cost and strong currency in Switzerland, which is probably not representative of what a well chosen external contractor could offer. Should another team wish to reproduce the technique, we are happy to provide more technical details on our experience upon request, so as to reduce the cost and time.
10 - This sentence has been reformulated.
11 - We thank the referee for pointing our this issues, which have been corrected in the new version of the paper.

Changes requested in report 1:

1- We have modified the figure following these recommendations.
2- We have included an appendix with the derivation of Eq 1
3 - Our discussion of the roles of conduction and convection as well as the Nusselt number was confusing. We have rewritten section 2 in order to clarify these discussions.
4 - The intended use of the coils is in pairs with a Helmholtz configuration, such that at the position of the atoms there is no gradient but a finite curvature. We have now included a new version of figure 3 where the simulated field distribution for this configuration is shown. We also state numbers concerning the field curvature in the text.
Concerning temporal variations due to thermal heating of the coils, we thank the referee for this insightful question. Section 4.2 now includes a paragraph describing the expected variation of magnetic field at fixed current, due to thermal expansion of the coil.
5 - Indeed titanium was primarily chosen for its non magnetic character. Titanium has similar electrochemical potential as stainless steel, and is more cathodic than copper. In our configuration,the surface of titanium exposed to water is very small. This is the recommended configuration for mitigating galvanic corrosion (see for exemple the standard MIL-STD-889C). We use deionized water for the cooling fluid, which further reduces the risk of corrosion.
6 - The temperature was not regulated, but we did not notice variations of the water temperature while acquiring the data.
7 - The relevant sentences have been corrected.

---

## Round 3 · Author Response

We are pleased to submit an updated version of the manuscript containing the revisions requested by the two referees.

---

## Round 3 · List of Changes

- Typos have been corrected
- Mentions of a 'Helmholtz' configuration for the coils has been removed.

---

## Editorial Decision

published